# Modified musculofascial lengthening technique for submuscular ulnar nerve transposition in cubital tunnel syndrome

Sang-Pil So[1], Won Sun Lee[2], KiHyeok Ku[3], Young Ho Shin[2], Jae Kwang Kim[2]*

1 Department of Orthopedic Surgery, Armed Forces Capital Hospital, Seongnam-si, Gyeonggi-do, South Korea, 2 Department of Orthopedic Surgery, Asan Medical Center, University of Ulsan College of Medicine, Songpa-gu, Seoul, Republic of Korea, 3 Department of Orthopedic Surgery, Kyung Hee University Hospital at Gangdong, College of Medicine, Kyung Hee University, Dongdaemun-gu, Seoul Republic of Korea

* orth4535@gmail.com

## Abstract

### Objective

Cubital tunnel syndrome is a common peripheral neuropathy of the upper extremity. Anterior transposition of the ulnar nerve is an established surgical treatment option for this condition. This study aimed to introduce a novel musculofascial lengthening technique that uses only a portion of the flexor-pronator muscle mass for submuscular anterior transposition of the ulnar nerve and investigate its clinical outcomes.

### Methods

We evaluated 28 patients (29 cases; 1 patient had bilateral involvement) diagnosed with cubital tunnel syndrome who were treated with surgical decompression and submuscular anterior transposition of the ulnar nerve using our novel technique. Mean follow-up was 19.1 months (range, 12–31). Patient-reported outcomes were assessed using the Boston Carpal Tunnel Questionnaire (BCTQ), Disabilities of the Arm, Shoulder, and Hand (DASH) Questionnaire, and numeric rating scale (NRS) for pain. Objective outcomes including light touch perception, static two-point discrimination, and grip strength were also assessed. Modified Bishop score and postoperative complications were also evaluated.

### Results

BCTQ symptom severity and functional status scores, DASH score, and NRS for pain score showed significant improvement after surgery. Light touch perception, static two-point discrimination, and grip strength also significantly improved after surgery. All patients showed excellent or good results according to the modified Bishop scoring system. No recurrence or complications occurred.

Data Availability Statement: All relevant data are within the manuscript and its Supporting Information files.

**Funding:** The author(s) received no specific funding for this work.

**Competing interests:** The authors have declared that no competing interests exist.

## Conclusion

Our novel musculofascial lengthening technique that uses only a portion of the flexor-pronator muscle mass for submuscular anterior transposition of the ulnar nerve reliably achieves good results with minimal complications in patients with cubital tunnel syndrome.

## Introduction

Cubital tunnel syndrome is a common peripheral neuropathy of the upper extremity that occurs because of ulnar nerve entrapment at the elbow [1, 2]. Simple decompression, anterior transposition of the ulnar nerve, and medial epicondylectomy are the established surgical treatment options for this condition [3–5]. Selection of surgical technique usually depends upon the surgeon's preference. Furthermore, the most effective surgical technique to treat this condition remains under debate [6].

Anterior transposition of the ulnar nerve is widely used to treat cubital tunnel syndrome and is classified according to the location of the ulnar nerve after transposition as follows: subcutaneous, intramuscular, or submuscular. A previous cadaveric biomechanical study that compared intraneural ulnar nerve pressure among surgical techniques demonstrated that the musculofascial lengthening technique for submuscular transposition reduces pressure in all degrees of elbow flexion [7]. This technique using the entire flexor-pronator muscle mass is useful to treat cubital tunnel syndrome [8, 9]. However, the classic musculofascial lengthening procedure in submuscular transposition requires extensive dissection, which is associated with disadvantages such as longer mean operative time, greater bleeding and postoperative pain, and higher rate of infection [10–13]. In addition, dissection of the entire flexor-pronator mass might delay the initiation of postoperative range of motion exercises, which may result in development of an elbow contracture [14].

To reduce these shortcomings, we devised a novel musculofascial lengthening technique that uses only a portion of the flexor-pronator mass for submuscular anterior transposition of the ulnar nerve. The primary objective of this study was to assess subjective and objective outcomes of this technique. Our secondary objective was to investigate patient satisfaction and complications.

## Materials and methods

### Patients

This study was approved by the institutional review board of our institution (2020–0955). Forty-nine cases of cubital tunnel syndrome diagnosed in 48 consecutive patients (one patient was affected bilaterally) who were treated surgically in our center from December 2016 to January 2020 were eligible for study inclusion. All patients underwent surgical decompression and anterior transposition of the ulnar nerve performed by a single orthopedic surgeon utilizing the novel technique. Diagnosis of cubital tunnel syndrome was based on clinical presentation and confirmed by electrodiagnostic testing for all included patients. Common manifestations were numbness or paresthesia in the areas innervated by the ulnar nerve, decreased grip strength, and intrinsic hand muscle atrophy. Surgical indications were objective sensory-motor changes and persistent symptoms after six months of conservative treatment including lifestyle modification, night extension brace and medication. Patients who were scheduled to undergo concurrent surgery of the same extremity, those who had undergone

previous surgery of the same elbow, and those who did not complete at least one year of follow-up were excluded. One patient had concurrent carpal tunnel release of the same extremity and 11 patients underwent revision surgery because of failed simple decompression of the ulnar nerve. Follow-up was under one year in eight patients. Therefore, 28 patients (29 cases) were finally included for analysis.

## Surgical technique

Surgery was performed with the patient in the supine position under regional anesthesia. After inflation of the tourniquet, a curved longitudinal incision was made 1 cm posterior to the medial epicondyle. The subcutaneous layer and fascia were carefully dissected with iris scissors to avoid damage to the medial antebrachial cutaneous nerve. The medial intermuscular septum and arcade of Struthers were released proximally, followed by release of Osborne's ligament and the aponeurosis of flexor carpi ulnaris (Fig 1). Branch of ulnar nerve to the flexor carpi ulnaris was released. Then, the ulnar nerve was then freely mobilized and the subcutaneous layer above the flexor-pronator fascia was dissected. A Z-shaped outline was drawn on the flexor-pronator fascia (Fig 2). A proximal flap, using the fascial layer of the middle third of the flexor-pronator mass, and distal flap, using the fascial and muscular layer of the humeral head of the flexor carpi ulnaris, were then created (Fig 3) and lengthened together by suturing, taking care not to injure the medial collateral ligament. Musculofascial lengthening was performed loosely; not to tent or compress the transposed ulnar nerve. Tension and degree of kinking of the ulnar nerve were evaluated by flexing and extending the elbow (Fig 4) and sutures were applied. The tourniquet was then deflated, and meticulous hemostasis was achieved. The wound was then irrigated and closed in layers. Postoperatively, a long arm YOGIPS splint was applied, and elbow range of motion exercises were started two days after

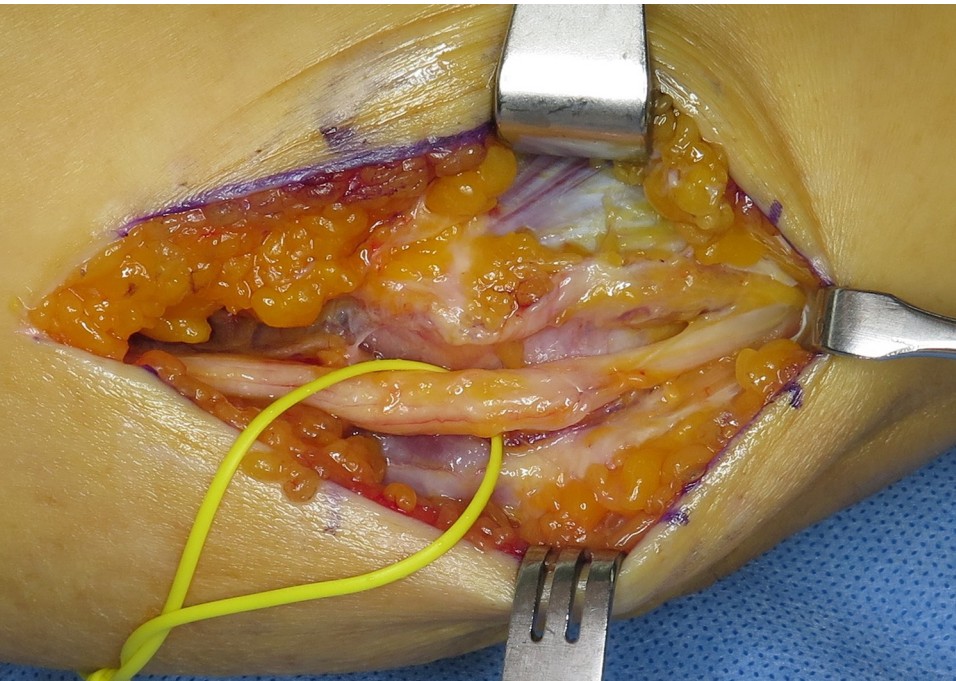

**Fig 1. Intraoperative photograph after release of Osborne's ligament and the aponeurosis of the flexor carpi ulnaris.** The ulnar nerve was protected with a yellow vessel loop.

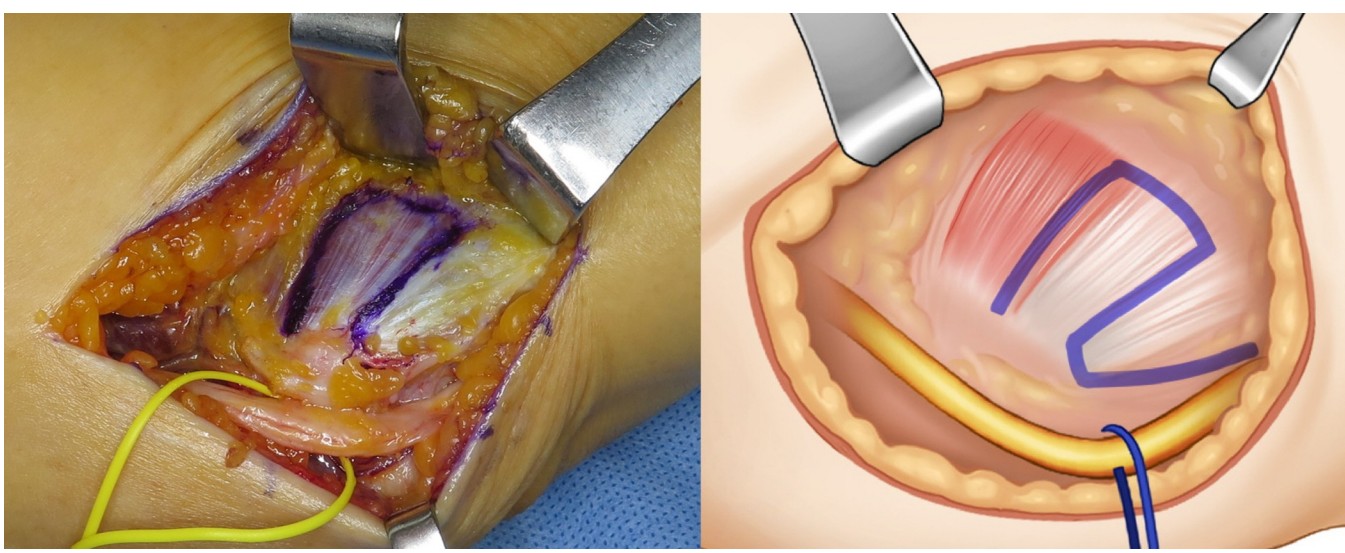

**Fig 2. Intraoperative photograph and illustration image of the Z-shaped outline drawn on the flexor-pronator fascia and muscle mass.**

surgery. Patients were instructed to wear the splint at all times except when performing exercises two or three times a day. Two weeks after surgery, the sutures and splint were removed without further restriction.

## Clinical evaluations

One examiner, independent of treating surgeons, performed all the testing and distributed the questionnaires.

We collected patient-reported outcomes using the Boston Carpal Tunnel Questionnaire (BCTQ), Disabilities of the Arm, Shoulder and Hand (DASH) questionnaire, and numeric rating scale (NRS) for pain preoperatively and at final follow-up. The BCTQ [15] is composed of symptom severity and functional status scales; the symptom severity scale consists of 11

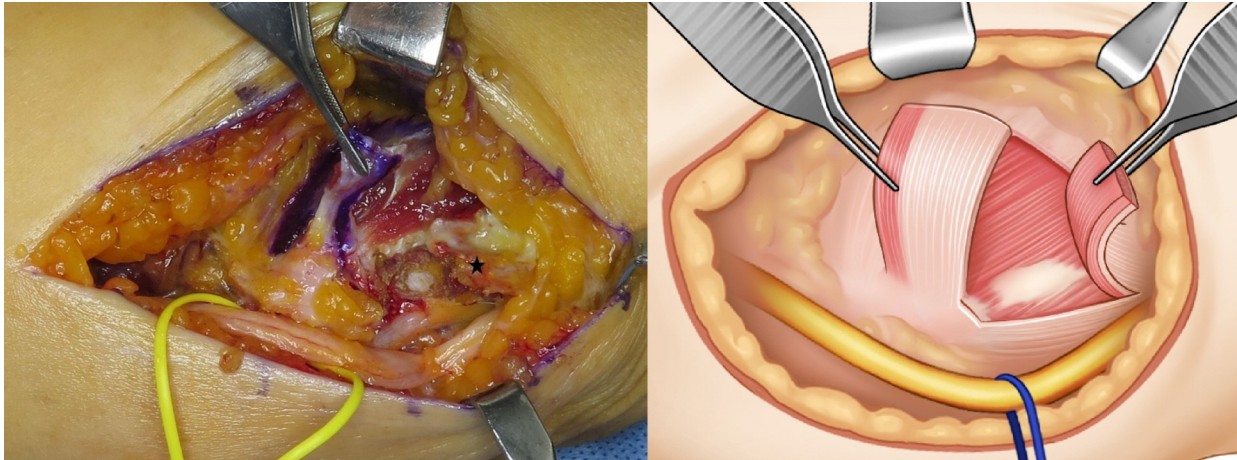

**Fig 3. Intraoperative photograph and illustration of a proximal flap (fascial layer of the middle third of the flexor-pronator mass) and a distal flap (fascial and muscular layers of the humeral head of the flexor carpi ulnaris).** The intraoperative photograph shows a forceps holding the proximal flap. The distal flap is indicated by the star. The illustration shows forceps holding both the proximal and distal flaps.

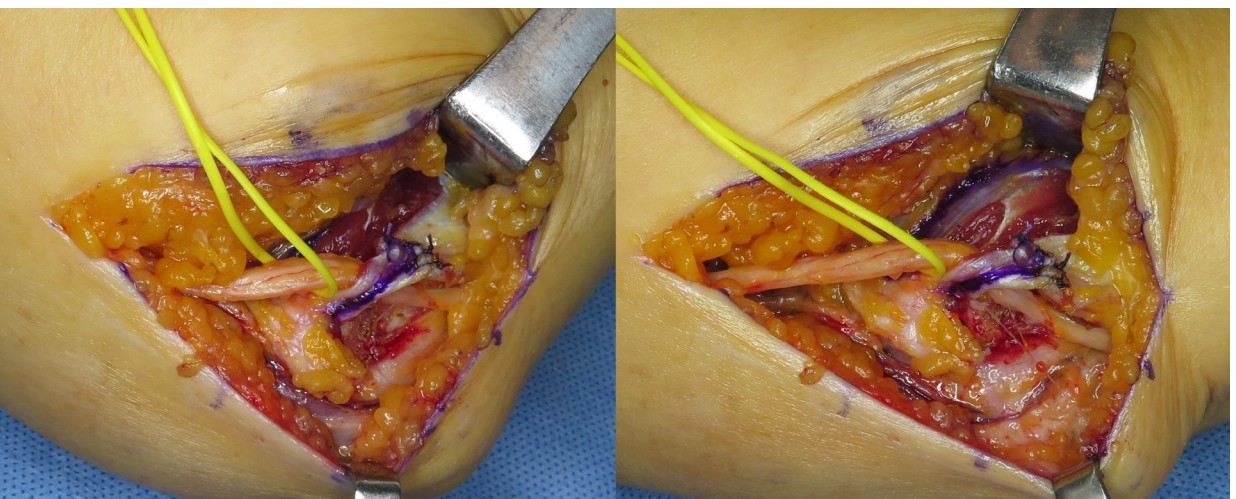

**Fig 4. Intraoperative photograph after musculofascial lengthening by suturing.** Tension and degree of kinking of the ulnar nerve were evaluated by flexing and extending the elbow.

questions about the degree of pain, numbness, weakness, and loss of dexterity, while the functional status scale consists of eight questions about difficulties in performing daily tasks. Each question has five possible responses ranging from 1 (no symptoms) to 5 (severe symptoms) and the average score was calculated for each question for the analysis. The DASH questionnaire [16] consists of 30 items about the degree of difficulty when carrying out different physical activities, severity of symptoms such as pain, weakness, and stiffness, and their impact on psychosocial functioning. The scores for each item are added and the final score is analyzed using a scale ranging from 0 (no disability) to 100 (most severe disability). The NRS for pain in the affected arm was assessed on a scale of 0 (no pain) to 10 (worst imaginable pain).

Objective sensory-motor functions, including light touch perception, static two-point discrimination, and grip strength, were also assessed preoperatively and at final follow-up. Light touch perception was assessed using the Semmes-Weinstein monofilament test. This test measures touch with the Touch Test Sensory Evaluator (North Coast Medical Inc., Morgan Hill, CA, USA) using five probes and was graded from 0 to 5, according to the procedure described by Bell-Krotoski (Table 1) [17].

Static two-point discrimination was measured at the distal phalanx of the little finger using the Baseline® Two Point Discriminator (Fabrication Enterprises, White Plains, NY, USA) and was graded from 0 to 4, according to the referred values for palm and fingers of the hand source set by the American Society of Hand Therapists (Table 2) [18]. Grip strength was measured in both extremities at the same time using a handgrip dynamometer (TKK5401, TAKEI corporation, Niigata-city, Japan).

**Table 1. Grade and interpretation of the Semmes-Weinstein monofilament test.**

| Grade | Interpretation | Monofilament size |
|---|---|---|
| 5 | Normal | 2.83 |
| 4 | Diminished light touch | 3.61 |
| 3 | Diminished protective sensation | 4.31 |
| 2 | Loss of protective sensation | 4.56 |
| 1 | Deep pressure sensation only | 6.65 |
| 0 | Loss of sensation | Unresponsive to 6.65 |

**Table 2. Grade and interpretation of the static two-point discrimination test.**

| Grade | Interpretation | Two-point discrimination (mm) |
|---|---|---|
| 4 | Normal | <6 |
| 3 | Fair | 6–10 |
| 2 | Poor | 11–15 |
| 1 | Protective | One point perceived |
| 0 | Anesthesia | No points perceived |

We also evaluated the modified Bishop score [9, 19, 20] at the final follow-up. The modified Bishop scoring system consists of seven categories; satisfaction with surgery, improvement after surgery, severity of residual symptoms, working status, leisure activity, intrinsic muscle strength, and static two-point discrimination (Table 3). Scores of each category were then added together and classified as excellent (score 8 and above), good (score 5 to 7), fair (score 3 and 4), or poor (score below 2). Recurrence of cubital tunnel syndrome and surgical complications such as hematoma, surgical site infection, injury of the medial antebrachial cutaneous nerve, and elbow contracture were determined by medical record review.

**Table 3. Modified Bishop scoring system.**

| | Points |
|---|---|
| Satisfaction | |
| Satisfied | 2 |
| Satisfied with reservation | 1 |
| Dissatisfied | 0 |
| Improvement | |
| Better | 2 |
| Unchanged | 1 |
| Worse | 0 |
| Severity | |
| Asymptomatic | 3 |
| Mild, occasional | 2 |
| Moderate | 1 |
| Severe | 0 |
| Work status | |
| Working or able to work at previous job | 1 |
| Not working because of ulnar neuropathy | 0 |
| Leisure activity | |
| Unlimited | 1 |
| Limited | 0 |
| Strength | |
| Intrinsic muscle strength normal (M5) | 2 |
| Intrinsic muscle strength reduced to M4 | 1 |
| Intrinsic muscle strength less than or equal to M3 | 0 |
| Sensibility (static two-point discrimination) | |
| ≤6 mm | 1 |
| >6 mm | 0 |
| Total | 12 |

M, Medical council grading system

## Statistical analysis

Continuous variables are presented as means with 95% confidence interval (CI). Ordinal variables are reported as numbers of patients. Pre- and postoperative continuous and ordinal variables were compared using the Wilcoxon signed rank test. $P < 0.05$ was considered significant. Statistical analyses were performed using SPSS software version 21 (IBM Corp., Armonk, NY, USA).

## Results

Mean patient age was 59.9 years (range, 18–77) and mean follow-up was 19.1 months (range, 12–31). All patients were right-handed. Seventeen cases were performed on the right side.

The BCTQ symptom severity score significantly improved from 2.3 (95% CI, 1.9–2.6) to 1.7 (95% CI, 1.5–1.8) ($P <$0.001). The BCTQ functional status score significantly improved from 2.0 (95% CI, 1.6–2.3) to 1.4 (95% CI, 1.2–1.6) ($P = 0.005$). The DASH score significantly improved from 24.1 (95% CI, 16.2–31.9) to 10.6 (95% CI, 6.2–15.0) ($P <$0.001). The NRS score for pain significantly improved from 3.8 (95% CI, 2.8–4.8) to 2.0 (95% CI, 1.3–2.7) ($P = 0.002$).

Light touch perception measured by the Semmes-Weinstein monofilament test significantly improved after surgery ($P <$0.001). The number of patients presenting with grades 3 and 4 increased, while those with grades 0, 1, and 2 decreased (Table 4). Static two-point discrimination also showed significant improvement postoperatively ($P = 0.002$). The number of patients with grade 4 increased and those with grades 1, 2 and 3 decreased (Table 4). Grip strength significantly improved from 21.4 Kg. (95% CI, 18.3–24.4) to 26.2 Kg. (95% CI, 23.2–29.0) ($P = 0.001$).

All cases showed excellent or good results as assessed by the modified Bishop scoring system: 25 patients reported excellent results while three reported good results. Table 5 shows the number of patients according to the modified Bishop score. Any complications during perioperative period and follow-up were not recorded.

**Table 4. Results of examinations of objective sensory-motor function.**

|  | Preoperative (n) | Postoperative (n) | *P*-value |
|---|---|---|---|
| Light touch perception |  |  | <0.01 |
| Normal | 1 | 1 |  |
| Diminished light touch | 11 | 16 |  |
| Diminished protective sensation | 8 | 12 |  |
| Loss of protective sensation | 2 | 0 |  |
| Deep pressure sensation only | 4 | 0 |  |
| Loss of sensation | 3 | 0 |  |
| Static two-point discrimination |  |  | <0.01 |
| Normal | 6 | 19 |  |
| Fair | 14 | 9 |  |
| Poor | 4 | 0 |  |
| Protective | 5 | 1 |  |
| Anesthesia | 0 | 0 |  |
| First dorsal interosseus muscle atrophy |  |  | <0.01 |
| Present | 17 | 8 |  |
| Absent | 12 | 21 |  |

**Table 5. Number of patients according to modified Bishop score.**

| Modified Bishop score | Patients (n) |
|---|---|
| 12 | 2 |
| 11 | 8 |
| 10 | 4 |
| 9 | 5 |
| 8 | 6 |
| 7 | 4 |
| below 6 | 0 |

## Discussion

Simple decompression, anterior transposition of the ulnar nerve (subcutaneous, intramuscular, submuscular), and medial epicondylectomy are known surgical options for the cubital tunnel syndrome [21, 22] and this study demonstrated favorable outcomes of submuscular anterior transposition of the ulnar nerve using a musculofascial lengthening technique that uses only a portion of the flexor-pronator mass. Significant improvement occurred in both subjective patient-reported outcomes and objectively measured functions. All cases showed excellent or good results according to the modified Bishop scoring system.

Disadvantages and postoperative complications have been reported for the various cubital tunnel syndrome treatment procedures. Although simple decompression including utilizing minimal invasive endoscopic techniques has small skin incision with less vascular insult to the nerve and resulting in faster recovery of the patient, simple decompression is associated with a higher recurrence rate after surgery because it can cause postoperative ulnar nerve subluxation and is unable to reduce the intraneural pressure caused by traction of the ulnar nerve during elbow flexion [7, 23–29]. Medial epicondylectomy can cause severe postoperative pain, flexor-pronator weakness, and valgus instability if more than 40% of the medial epicondyle is removed [30, 31]. In contrast, anterior transposition eliminates the natural and pathological traction and compression forces during elbow flexion [32]. However, subcutaneous transposition results in significantly higher postoperative intraneural pressure in complete elbow extension [7]. In addition, due to its superficial position after subcutaneous transposition, the ulnar nerve can be hypersensitive after surgery and it has been suggested that thin patients might be prone to repeated trauma to the transposed ulnar nerve [33]. And, subcutaneous transposition displayed more perineural scar tissue and unhealthy axons [34].

Submuscular transposition has the following advantages: it creates the straightest path for the nerve and provides a vascularized muscular bed [13, 35, 36]. Dellon and Coert introduced the musculofascial lengthening technique for submuscular transposition using the flexor-pronator fascia as a proximal flap and the entire flexor-pronator muscle mass as a distal flap and reported good to excellent results according to their own criteria in 88% of cases [8]. Nouhan and Kleinert introduced a musculofascial lengthening technique that used the entire flexor-pronator mass and Z-plasty; of their 33 reported cases, 12 showed excellent results and 20 showed good results according to the modified Bishop scoring system [9]. However, these techniques require extensive dissection of the entire flexor-pronator mass, which can cause several complications [10–13].

In this study, instead of using the entire flexor-pronator muscle mass as the distal flap, only the muscular layer of the humeral head of the flexor carpi ulnaris was dissected and lengthened together with the fascial layer of the middle third of the flexor-pronator mass to form a bed for the ulnar nerve. Several advantages are expected by using only a portion of the flexor-pronator

muscle mass. After using the traditional musculofascial lengthening technique for submuscular transposition, Nouhan and Kleinert reported a 10% decrease in grip strength in patients who had normal preoperative grip strength and Novak et al. reported that 21% of patients had less grip strength after surgery [9, 37]. In this study, mean grip strength improved on the affected side from 21.4 Kg. to 26.2 Kg. (a 22% increase). In addition, we allowed the patients to perform range of motion exercises just two days after surgery, which might have prevented elbow stiffness and contracture. Furthermore, owing to its relative simplicity, partial dissection of the flexor-pronator muscle mass is expected to result in shorter operative time, less bleeding and postoperative pain.

This study has several limitations. First, it was retrospective in design and had a relatively short follow-up period and small study population. Second, we did not analyze a control group. Third, we did not conduct specific physical examinations to assess the function of the intrinsic muscles innervated by the ulnar nerve. Finally, selection bias may have been introduced because we excluded patients with less than one year of follow-up. Nonetheless, this study also has several strengths. It reports the results of a single surgeon's consistent surgical technique and postoperative care. Furthermore, this study measured various aspects of patient data ranging from questionnaires reflecting clinical status to several examinations of sensory-motor function. Evaluation of various aspects enhances the reliability of our overall results.

These results suggest that our modified musculofascial lengthening technique that uses only a portion of the flexor-pronator muscle mass for submuscular anterior transposition of the ulnar nerve reliably achieves good results with minimal complications in patients with cubital tunnel syndrome.

## Supporting information

**S1 Dataset. Raw data of patient demographics and preoperative and postoperative assessments.** This dataset contains anonymized raw data collected from all patients included in the study. The data encompass:

- **Patient Demographics**:
  - Patient number (anonymous identifier)
  - Sex
  - Age
  - Affected side (right or left)
  - Dominant hand
- **Surgical Details**:
  - Operation date
- **Preoperative Assessments**:
  - Evaluation date
  - Boston Carpal Tunnel Questionnaire (BCTQ):
    - Symptom Severity Score
    - Functional Status Score
  - Disabilities of the Arm, Shoulder, and Hand (DASH) Score

○ Presence of thenar atrophy

○ Pain Numeric Rating Scale (NRS) Score

○ Grip Strength measurement

○ Two-Point Discrimination test results

○ Light Touch Perception test results

- **Postoperative Assessments**:

  ○ Follow-up date

  ○ BCTQ:

    ■ Symptom Severity Score

    ■ Functional Status Score

  ○ DASH Score

  ○ Presence of thenar atrophy

  ○ Pain NRS Score

  ○ Patient satisfaction for surgery

  ○ Grip Strength measurement

  ○ Two-Point Discrimination test results

  ○ Light Touch Perception test results
  **Note**: All data have been de-identified to ensure patient confidentiality.
  (XLSX)

## Author Contributions

**Conceptualization:** Jae Kwang Kim.

**Data curation:** Won Sun Lee, KiHyeok Ku.

**Writing – original draft:** Sang-Pil So.

**Writing – review & editing:** Young Ho Shin, Jae Kwang Kim.

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
