## [Decision Letter · Decision Letter 0]

21 Aug 2024

PONE-D-24-12479Modified musculofascial lengthening technique for submuscular ulnar nerve transposition in cubital tunnel syndromePLOS ONE

Dear Dr. Kim,

Thank you for submitting your manuscript to PLOS ONE. After careful consideration, we feel that it has merit but does not fully meet PLOS ONE’s publication criteria as it currently stands. Therefore, we invite you to submit a revised version of the manuscript that addresses the points raised during the review process.

We look forward to receiving your revised manuscript.

Kind regards,

Vanessa Carels

Staff Editor

PLOS ONE

Reviewers' comments:

Reviewer's Responses to Questions

**Comments to the Author**

1. Is the manuscript technically sound, and do the data support the conclusions?

Reviewer #1: Yes

Reviewer #2: Yes

2. Has the statistical analysis been performed appropriately and rigorously? 

Reviewer #1: Yes

Reviewer #2: I Don't Know

3. Have the authors made all data underlying the findings in their manuscript fully available?

Reviewer #1: Yes

Reviewer #2: Yes

4. Is the manuscript presented in an intelligible fashion and written in standard English?

Reviewer #1: Yes

Reviewer #2: Yes

5. Review Comments to the Author

Reviewer #1: Thank you for your effort on the study.

My comments:

The introduction section is well-written.

Materials and methods section:

In surgical technique, please give data about if there was any restriction after 2 post-operative 2 weeks.

What was your follow-up protocol after two weeks?

What was the timing of post-operative clinical assessment? Did you recall the patients for a final follow-up or you assess the patients in rutin follow-up?

If you have the clinical assessments in post-operative 3 and/or 6 months, it could be better adding them to see progression of clinical status. If you have that data please add it.

The results section is well-written.

Discussion section:

In the first paragraph before reporting most important finding of your study, please give a brief information about ulnar nerve entrapment treatments.

In this section, the most of the articles you cited are very old. You can site newer articles like:

-Lanzetti RM, Astone A, Pace V, D'Abbondanza L, Braghiroli L, Lupariello D, Altissimi M, Vadalà A, Spoliti M, Topa D, Perugia D, Caraffa A. Neurolysis versus anterior transposition of the ulnar nerve in cubital tunnel syndrome: a 12 years single secondary specialist centre experience. Musculoskelet Surg. 2021 Apr;105(1):69-74

-Byun YS, Lee SU, Park IJ, Im JH, Hong SA. Comparison of in-situ release and submuscular anterior transposition of ulnar nerve for refractory cubital tunnel syndrome, previously treated with subfascial anterior transfer-A retrospective study of 24 cases. Injury. 2023 Dec;54(12):111061.

-Liu CH, Wu SQ, Ke XB, Wang HL, Chen CX, Lai ZL, Zhuang ZY, Wu ZQ, Lin Q. Subcutaneous Versus Submuscular Anterior Transposition of the Ulnar Nerve for Cubital Tunnel Syndrome: A Systematic Review and Meta-Analysis of Randomized Controlled Trials and Observational Studies. Medicine (Baltimore). 2015 Jul;94(29):e1207.

-Ergen E, Ertem K, Karakaplan M, Kavak H, Aslantürk O. Review of Anterior Submuscular Transposition of Ulnar Nerve for Cubital Tunnel Syndrome. Niger J Clin Pract. 2021 Aug;24(8):1170-1173.

Also there is no discussion about minimal invasive endoscopic techniques, it could be better to add one paragraph about that topic.

-Morse LP, McGuire DT, Bain GI. Endoscopic ulnar nerve release and transposition. Tech Hand Up Extrem Surg. 2014 Mar;18(1):10-4.

-Marcheix PS, Vergnenegre G, Chevalier C, Hardy J, Charissoux JL, Mabit C. Endoscopic ulnar nerve release at the elbow: Indications and outcomes. Orthop Traumatol Surg Res. 2016 Feb;102(1):41-5.

-Smeraglia F, Del Buono A, Maffulli N. Endoscopic cubital tunnel release: a systematic review. Br Med Bull. 2015;116:155-63.

Reviewer #2: This retrospective study evaluates 29 cubital tunnel syndrome operations using the flexor-pronator musculofascial lengthening technique combined with anterior submuscular transposition of the ulnar nerve. The authors focus on reporting the outcomes of this method.

Introduction: Well-written.

Materials and Methods:

- Lines 82-84, ‘Surgical indications included objective sensory-motor changes and persistent symptoms after six months of conservative treatment.’ Do you perform electrodiagnostic examinations, such as electromyography (EMG) and nerve conduction studies (NCS), for these patients with cubital tunnel syndrome? Additionally, what does your conservative treatment entail?

- Lines 95-96, ‘The medial intermuscular septum and arcade of Struthers were released proximally.’ How do you release the medial intermuscular septum, or do you resect it?

- For the surgical procedures: Do you release the branch(es) of the ulnar nerve to the flexor carpi ulnaris (FCU) muscle to avoid kinking during anterior transposition? In Figure 3, it appears that the fascial septum between the flexor carpi ulnaris and the flexor-pronator muscles is very obvious after the musculofascial flap is harvested. Could this cause tenting and compression of the transposed ulnar nerve?

- Line 146, 'the fifth finger' should be replaced with 'the little finger' or 'the small finger'.

Results and discussion

- In your study, the mean grip strength on the operated side improved from 10.7 kg to 13.1 kg. It appears that the grip strength is generally lower than that of the average population and what is reported in the literature. Could there be any reason for this?

- In addition to observing improvements in intrinsic muscle atrophy, did you perform any physical examinations to test the function of the intrinsic muscles?

- As you mentioned in the introduction (lines 54-56), submuscular transposition reduces intraneural ulnar nerve pressure. In your method, will the area of the proximal flap, the proximal-based fascial layer of the middle third of the flexor-pronator mass, cause increased intraneural ulnar nerve pressure, since this part involves only subfascial nerve transfer?

Conclusion: I would consider this a modified musculofascial lengthening technique rather than a novel one.

6. PLOS authors have the option to publish the peer review history of their article (what does this mean?). If published, this will include your full peer review and any attached files.

Reviewer #1: No

Reviewer #2: No

---

## [Author Response · Author response to Decision Letter 0]

7 Oct 2024

Response to REVIEWER 1 comment

We would like to thank you for the opportunity to resubmit our revised manuscript. We would also like to express our gratitude to the reviewers for their positive feedback and helpful comments. The manuscript has benefited immensely from these insightful suggestions. We have improved the article based on the critiques and suggestions. Please, see below our comments to the critiques and our revised manuscript. We have provided point-by-point responses to each suggestion below. The changes are also highlighted in the revised manuscript. 

Reviewer #1: Thank you for your effort on the study.

My comments:

The introduction section is well-written.

Materials and methods section:

In surgical technique, please give data about if there was any restriction after 2 post-operative 2 weeks.

Page 4 (Lines 90): Following your comments, we added information about further restriction after postoperative two weeks (no further restriction). 

What was your follow-up protocol after two weeks? 

We tried to follow-up the patients at postoperative two weeks, three months, 1 year, 2 years and 3 years. 

What was the timing of post-operative clinical assessment? Did you recall the patients for a final follow-up or you assess the patients in rutin follow-up? If you have the clinical assessments in post-operative 3 and/or 6 months, it could be better adding them to see progression of clinical status. If you have that data please add it.

We tried to follow-up patients as mentioned above, however, the number of patients followed-up at each time point was restricted and main focus point of this study was clinical outcomes at the final follow-up; therefore, the clinical assessment data at the final follow-up was utilized for the analysis of this study.

The results section is well-written.

Discussion section:

In the first paragraph before reporting most important finding of your study, please give a brief information about ulnar nerve entrapment treatments.

Page 7 (Lines 155-157): Following your comments, we added possible surgical options for the cubital tunnel syndrome before reporting findings of our study.

In this section, the most of the articles you cited are very old. You can site newer articles like:

-Lanzetti RM, Astone A, Pace V, D'Abbondanza L, Braghiroli L, Lupariello D, Altissimi M, Vadalà A, Spoliti M, Topa D, Perugia D, Caraffa A. Neurolysis versus anterior transposition of the ulnar nerve in cubital tunnel syndrome: a 12 years single secondary specialist centre experience. Musculoskelet Surg. 2021 Apr;105(1):69-74

-Byun YS, Lee SU, Park IJ, Im JH, Hong SA. Comparison of in-situ release and submuscular anterior transposition of ulnar nerve for refractory cubital tunnel syndrome, previously treated with subfascial anterior transfer-A retrospective study of 24 cases. Injury. 2023 Dec;54(12):111061.

-Liu CH, Wu SQ, Ke XB, Wang HL, Chen CX, Lai ZL, Zhuang ZY, Wu ZQ, Lin Q. Subcutaneous Versus Submuscular Anterior Transposition of the Ulnar Nerve for Cubital Tunnel Syndrome: A Systematic Review and Meta-Analysis of Randomized Controlled Trials and Observational Studies. Medicine (Baltimore). 2015 Jul;94(29):e1207.

-Ergen E, Ertem K, Karakaplan M, Kavak H, Aslantürk O. Review of Anterior Submuscular Transposition of Ulnar Nerve for Cubital Tunnel Syndrome. Niger J Clin Pract. 2021 Aug;24(8):1170-1173.

Page 7(Lines 157), Page 8 (Lines 175,177): we added newer articles mentioned above in our discussion section, following your comments.

Also there is no discussion about minimal invasive endoscopic techniques, it could be better to add one paragraph about that topic.

-Morse LP, McGuire DT, Bain GI. Endoscopic ulnar nerve release and transposition. Tech Hand Up Extrem Surg. 2014 Mar;18(1):10-4.

-Marcheix PS, Vergnenegre G, Chevalier C, Hardy J, Charissoux JL, Mabit C. Endoscopic ulnar nerve release at the elbow: Indications and outcomes. Orthop Traumatol Surg Res. 2016 Feb;102(1):41-5.

-Smeraglia F, Del Buono A, Maffulli N. Endoscopic cubital tunnel release: a systematic review. Br Med Bull. 2015;116:155-63.

Page 7 (Lines 167-168): we added a brief introduction regarding minimal invasive endoscopic release techniques in the discussion section following your comments.

Response to REVIEWER 2 comment

We would like to thank you for the opportunity to resubmit our revised manuscript. We would also like to express our gratitude to the reviewers for their positive feedback and helpful comments. The manuscript has benefited immensely from these insightful suggestions. We have improved the article based on the critiques and suggestions. Please, see below our comments to the critiques and our revised manuscript. We have provided point-by-point responses to each suggestion below. The changes are also highlighted in the revised manuscript. 

Reviewer #2: This retrospective study evaluates 29 cubital tunnel syndrome operations using the flexor-pronator musculofascial lengthening technique combined with anterior submuscular transposition of the ulnar nerve. The authors focus on reporting the outcomes of this method.

Introduction: Well-written.

Materials and Methods:

- Lines 82-84, ‘Surgical indications included objective sensory-motor changes and persistent symptoms after six months of conservative treatment.’ Do you perform electrodiagnostic examinations, such as electromyography (EMG) and nerve conduction studies (NCS), for these patients with cubital tunnel syndrome? 

Page 3 (Lines 60): Following your comments, we added “all included patients” to underline that we confirmed diagnosis (cubital tunnel syndrome) by EMG and NCS for all included patients. Conservative treatment included lifestyle modification, night extension splint and medication. 

Additionally, what does your conservative treatment entail?

Page 3 (Lines 63): We added detailed information of conservative treatment at the manuscript following your comments. Conservative treatment included lifestyle modification, night extension splint and medication.

- Lines 95-96, ‘The medial intermuscular septum and arcade of Struthers were released proximally.’ How do you release the medial intermuscular septum, or do you resect it? 

We simply released the medial intermuscular septum without further resection.

- For the surgical procedures: Do you release the branch(es) of the ulnar nerve to the flexor carpi ulnaris (FCU) muscle to avoid kinking during anterior transposition? In Figure 3, it appears that the fascial septum between the flexor carpi ulnaris and the flexor-pronator muscles is very obvious after the musculofascial flap is harvested. Could this cause tenting and compression of the transposed ulnar nerve? 

Page 4 (Line 77-78, 83-84): we added further detail of surgical procedure, following your comments. We release the FCU muscle branch of the ulnar nerve to prevent kinking of the ulnar nerve and therefore leading to the ease of anterior transposition. And, musculofascial lengthening was performed loosely; therefore not to tent or compress the transposed ulnar nerve. 

- Line 146, 'the fifth finger' should be replaced with 'the little finger' or 'the small finger'.

Page 5 (Lines 113): We replaced “the fifth finger” to “little finger”, following your comments.

Results and discussion

- In your study, the mean grip strength on the operated side improved from 10.7 kg to 13.1 kg. It appears that the grip strength is generally lower than that of the average population and what is reported in the literature. Could there be any reason for this?

Page 6 (Line 147-148), Page 8 (Lines 192-193): After your comment, we reviewed patients’ hand grip strength data again. We found that grip strength was originally measured by “kilograms”, however, there was error during data acquisition. Data was acquired and analyzed as grip strength measured by “pounds”; therefore, overall decrease of grip strength was made after conversion of unit (from pounds to kilograms). We re-analyzed grip strength data by “kilograms” and edited results and discussion section. Further, we added accurate P-values for each analysis.

- In addition to observing improvements in intrinsic muscle atrophy, did you perform any physical examinations to test the function of the intrinsic muscles?

In our study, we did not perform specific physical examinations to test the function of the intrinsic muscles. While there are clinical methods to assess intrinsic muscle strength, such as manual muscle testing, these methods can be subjective and may lack the sensitivity needed for quantitative analysis in a research setting. We focused on objective measurements like grip strength, light touch perception, and static two-point discrimination, which provided reliable and reproducible data for evaluating the outcomes of our surgical technique.

- As you mentioned in the introduction (lines 54-56), submuscular transposition reduces intraneural ulnar nerve pressure. In your method, will the area of the proximal flap, the proximal-based fascial layer of the middle third of the flexor-pronator mass, cause increased intraneural ulnar nerve pressure, since this part involves only subfascial nerve transfer?

As we mentioned at the materials and method section, musculofascial lengthening by loose suturing did not lead to increased intraneural ulnar nerve pressure.

Conclusion: I would consider this a modified musculofascial lengthening technique rather than a novel one.

Page 8 (Line 205): We changed the phrase “novel musculofascial lengthening technique” to “modified musculofascial lengthening technique” following your comments.

---

## [Decision Letter · Decision Letter 1]

17 Oct 2024

PONE-D-24-12479R1Modified musculofascial lengthening technique for submuscular ulnar nerve transposition in cubital tunnel syndromePLOS ONE

Dear Dr. Kim,

Thank you for submitting your manuscript to PLOS ONE. After careful consideration, we feel that it has merit but does not fully meet PLOS ONE’s publication criteria as it currently stands. Therefore, we invite you to submit a revised version of the manuscript that addresses the points raised during the review process.

**ACADEMIC EDITOR: **Dear Authors, your R1 manuscript version has been reviewed by one expert in the field that retrieved some minor issues you should consider during the revision process. Please submit your revised manuscript by Dec 01 2024 11:59PM. If you will need more time than this to complete your revisions, please reply to this message or contact the journal office at plosone@plos.org. Please include the following items when submitting your revised manuscript:A rebuttal letter that responds to each point raised by the academic editor and reviewer(s). You should upload this letter as a separate file labeled 'Response to Reviewers'.A marked-up copy of your manuscript that highlights changes made to the original version. You should upload this as a separate file labeled 'Revised Manuscript with Track Changes'.An unmarked version of your revised paper without tracked changes. You should upload this as a separate file labeled 'Manuscript'.If applicable, we recommend that you deposit your laboratory protocols in protocols.io to enhance the reproducibility of your results. Protocols.io assigns your protocol its own identifier (DOI) so that it can be cited independently in the future. For instructions see: https://journals.plos.org/plosone/s/submission-guidelines#loc-laboratory-protocols. Additionally, PLOS ONE offers an option for publishing peer-reviewed Lab Protocol articles, which describe protocols hosted on protocols.io. Read more information on sharing protocols at https://plos.org/protocols?utm_medium=editorial-email&utm_source=authorletters&utm_campaign=protocols.

We look forward to receiving your revised manuscript.

Kind regards,

Emiliano Cè

Academic Editor

PLOS ONE

Journal Requirements:

Reviewers' comments:

Reviewer's Responses to Questions

**Comments to the Author**

1. If the authors have adequately addressed your comments raised in a previous round of review and you feel that this manuscript is now acceptable for publication, you may indicate that here to bypass the “Comments to the Author” section, enter your conflict of interest statement in the “Confidential to Editor” section, and submit your "Accept" recommendation.

Reviewer #2: All comments have been addressed

2. Is the manuscript technically sound, and do the data support the conclusions?

Reviewer #2: Yes

3. Has the statistical analysis been performed appropriately and rigorously? 

Reviewer #2: I Don't Know

4. Have the authors made all data underlying the findings in their manuscript fully available?

Reviewer #2: Yes

5. Is the manuscript presented in an intelligible fashion and written in standard English?

Reviewer #2: Yes

6. Review Comments to the Author

Reviewer #2: The manuscript has undergone a thorough revision, and all questions and suggestions have been addressed well.

I would recommend, if possible, that the authors include the results of tests for ulnar nerve intrinsic muscle function before and after surgery to demonstrate any recovery of ulnar intrinsic muscle function, such as Froment's test, the Crossed Finger test, and Wartenberg's sign. If this is not feasible, the authors should mention in the limitations section the absence of complete data from intrinsic muscle testing to demonstrate postoperative improvement.

7. PLOS authors have the option to publish the peer review history of their article (what does this mean?). If published, this will include your full peer review and any attached files.

Reviewer #2: No

---

## [Author Response · Author response to Decision Letter 1]

1 Jan 2025

Reviewer #2: The manuscript has undergone a thorough revision, and all questions and suggestions have been addressed well.

I would recommend, if possible, that the authors include the results of tests for ulnar nerve intrinsic muscle function before and after surgery to demonstrate any recovery of ulnar intrinsic muscle function, such as Froment's test, the Crossed Finger test, and Wartenberg's sign. If this is not feasible, the authors should mention in the limitations section the absence of complete data from intrinsic muscle testing to demonstrate postoperative improvement.

Page 9 (Line 203-4): Thank you for your insightful suggestion. In our study, we did not perform specific physical examinations to test the function of the intrinsic muscles innervated by the ulnar nerve, such as Froment's sign, Crossed Finger test, and Wartenberg's sign. We acknowledge that this is a limitation of our study. We have now included a statement in the limitations section to address this point:

---

## [Decision Letter · Decision Letter 2]

14 Jan 2025

Modified musculofascial lengthening technique for submuscular ulnar nerve transposition in cubital tunnel syndrome

PONE-D-24-12479R2

Dear Dr. Kim,

We’re pleased to inform you that your manuscript has been judged scientifically suitable for publication and will be formally accepted for publication once it meets all outstanding technical requirements.

Kind regards,

Emiliano Cè, Ph.D.

Academic Editor

PLOS ONE

Additional Editor Comments (optional):

Reviewers' comments:

Reviewer's Responses to Questions

**Comments to the Author**

1. If the authors have adequately addressed your comments raised in a previous round of review and you feel that this manuscript is now acceptable for publication, you may indicate that here to bypass the “Comments to the Author” section, enter your conflict of interest statement in the “Confidential to Editor” section, and submit your "Accept" recommendation.

Reviewer #2: All comments have been addressed

2. Is the manuscript technically sound, and do the data support the conclusions?

Reviewer #2: Yes

3. Has the statistical analysis been performed appropriately and rigorously? 

Reviewer #2: I Don't Know

4. Have the authors made all data underlying the findings in their manuscript fully available?

Reviewer #2: Yes

5. Is the manuscript presented in an intelligible fashion and written in standard English?

Reviewer #2: Yes

6. Review Comments to the Author

Reviewer #2: The authors have made significant improvements to this manuscript after a series of revisions. I believe the manuscript is now qualified for acceptance.

7. PLOS authors have the option to publish the peer review history of their article (what does this mean?). If published, this will include your full peer review and any attached files.

Reviewer #2: No

---

## [Editor Report · Acceptance letter]

20 Jan 2025

PONE-D-24-12479R2 

PLOS ONE

Dear Dr. Kim, 

I'm pleased to inform you that your manuscript has been deemed suitable for publication in PLOS ONE. Congratulations! Your manuscript is now being handed over to our production team.

Kind regards, 

on behalf of

Prof. Emiliano Cè 

Academic Editor

PLOS ONE